# An automatic water-occluding device to enable laryngectomee participation in water activities

**Samantha K. Denning**[1], **Michael A. Valleau**[1], **William J. Pelowski**[1], **Claire M. Chaisson**[2], **Kelli E. Grimes**[1], **Byron D. Erath**[1]*

**1** Department of Mechanical and Aeronautical Engineering, Clarkson University, Potsdam, NY, United States of America, **2** David D. Reh School of Business, Clarkson University, Potsdam, NY, United States of America

* berath@clarkson.edu

## Abstract

Individuals with a laryngectomy face a host of challenges ranging from restricted vocal communication to significant lifestyle modifications associated with breathing through a stoma. Although there are significant mental and physical health benefits achieved by returning to recreational pursuits that were enjoyed pre-surgery, there can be significant obstacles in doing so. One particular challenge arises during participation in water activities (e.g, fishing, boating, etc.) where accidental submersion poses a significant risk of drowning. This manuscript describes a proof-of-concept device that protects the airway from accidental incursion of water into the airway during unanticipated submersion in water, thereby allowing laryngectomees to return to participation in water activities. The device is designed to be worn comfortably for long periods of time, while not interfering with the common methods of replacement speech that are utilized post-laryngectomy.

## Introduction

Verbal communication is a distinguishing human characteristic that is a fundamental component of daily activities [1], arising from fluid-structure-acoustic interactions in the larynx [2]. Disruption of normal voice production can have wide-ranging impacts on social engagement, productivity, and general well-being [3–5]. This typically occurs as a result of some underlying pathology. If the ability of the larynx to protect the airway becomes compromised, as may occur from laryngeal cancer [6], bilateral vocal fold paralysis [7], or intractable aspiration [8], surgical intervention via laryngotracheal separation [8], tracheoesophageal diversion [8], or a total laryngectomy [9] may be necessary. In all scenarios, the superior and inferior portions of the airway are surgically separated and the trachea is redirected through the anterior cervical area of the neck, creating a tracheostoma (i.e., stoma) [10]. Following surgery, individuals must adapt to a plethora of challenges resulting from the newly-created airway.

Because one of the functions of the larynx is speech production [11], laryngectomees must utilize an alternate form of communication. About 90% of individuals are able to successfully produce voice post-surgery [12] using one of three methods: esophageal speech, an electro

**Funding:** This work was supported by the National Science Foundation (NSF), Division of Chemical, Bioengineering, Environmental, and Transport Systems (CBET), under Grant No. 1510367, awarded to B.D.E. (https://www.nsf.gov/div/index.jsp?div=CBET) The funders had no role in study design, data collection and analysis, decision to publish, or preparation of the manuscript.

**Competing interests:** The authors have declared that no competing interests exist.

larynx, or tracheoesophageal speech. Esophageal speech involves the controlled ingestion and expiration of air in the esophagus. During expulsion, the airflow excites self-oscillation of the pharyngoesophageal segment (PES). This in turn modulates the airflow, which produces sound [13, 14]. This technique is difficult to master and is infrequently used [12, 13, 15]. An electrolarynx uses a driver that impacts a diaphragm to produce monotonic sound [16]. When the diaphragm is pressed against the neck, sound is transmitted into the oral cavity, allowing production of voice through the normal posturing of the oral cavity [16]. Tracheoesophageal speech, by far the most common form of speech remediation, involves placement of a one-way prosthesis valve that connects the trachea with the esophagus [17, 18]. To produce voice, an individual occludes the stoma and uses lung pressure to drive the flow of air from the trachea, through the prosthesis, and into the esophagus. As air passes through the pharyngoesophageal segment, the resulting vibration of the PES produces sound. Other methods for speech production are also available (e.g., pneumatic artificial larynx [19], pneumatic bionic voice [20]), but are less common.

A laryngectomy also necessitates lifestyle changes that permanently alter an individual's physical, psychological, social, emotional, and nutritional well-being, with myriad secondary influences on their functional status [4]. Aside from voiced speech, other abilities that are lost or diminished include swallowing, taste, and smell, with various medical devices designed to address these challenges [21, 22]. Increased mucus production in the trachea and pulmonary system due to inhalation of cold dry air through the tracheostoma is also a potentially fatal concern [23]. Heat and moisture exchangers, such as the Provox® HME, can be placed over the tracheostoma to regulate the moisture level of the air that is inhaled [24]. Extraction of mucus from the trachea is a daily requirement [23], where devices have been developed to aid in manual expectoration [25]. Laryngectomees also lose the function of natural air filtration that breathing through the nose provides, increasing the risk of inhalation of harmful particulates, with commercially available tracheostoma filters often utilized to protect the airway [26]. Finally, fluid incursion into the pulmonary system through the tracheostoma is a significant concern, to the extent that laryngectomees must hold a washcloth over the stoma or use a shower shield when bathing [27].

Due to these significant lifestyle modifications, quality of life (QoL) can be severely diminished in laryngectomees. Prior studies of QoL have shown that the psychosocial impact of living with a laryngectomy is burdensome, with depression, feelings of social isolation, and decreased body image and sexuality quite common [10]. Approximately 40% of patients with head and neck cancers were diagnosed with depression six months after treatment [28]. Methods for coping with depression include: returning to previous activities, developing new leisure activities, keeping physically fit and active, and reintegrating socially with family and friends [12, 24]. This can include participation in recreational activities.

In the United States, 87 million people participated in recreational water activities including; boating, kayaking, personal water craft, and fishing, in the year 2020 [29]. These activities are often enjoyed with friends or family, making them an attractive option for helping laryngectomees cope with the psychosocial impacts of surgery. Prior work has shown that the laryngectomee population desires to be active, with one study reporting that of 38 laryngectomees polled, 31 were involved in sport before surgery, and of the 26 patients who responded post-op, all wanted to participate in some form of sport, including the offered hydrotherapy program [30].

Recognizing the importance of participation in physical activities, there have been several devices specifically designed for laryngectomees to confront the challenges associated with doing so [31]. Although the risk of drowning is significantly increased in laryngectomized individuals due to the risk of direct water ingress into the pulmonary tract, water sports can be

beneficial in the rehabilitation and improvement of QoL [32]. To this end, assistive devices have been developed to enable intentional/anticipated submersion of the user in water, such as while swimming. Examples include the Heimomed AQUA-THER™, Servona Servoaqua, Fahl Larchel®, the Larkel, and the Lary Freedom Snorkel® [33]. These devices attach to, and seal around, the stoma, usually via a tube that protrudes into the stoma. This same tube then connects the stoma to the mouth. By placing the end of the tube in the laryngectomee's mouth, the user can occlude water from entering the stoma while still breathing through their nose, which then routes the incoming air through the nasal passage, into the mouth, through the tube, and into the pulmonary tract. In this manner, the individual can hold their breath when submerged, and then breathe through their nose with their mouth closed when they surface. These devices are only functional, however, when the user can anticipate immersion under water. Furthermore, most do not enable replacement speech while wearing the device, requiring complete removal to allow communication, which can be cumbersome. Lacking, is a solution that allows an individual to participate in water activities where accidental submersion in water would likely result in drowning. The potentially fatal consequences associated with these recreational water activities often results in laryngectomees eschewing them, despite pre-surgical participation/enjoyment, and/or desires to do so. Note, because risk varies by activity, the current approach focuses on providing protection when participating in activities where risk arises due to slipping and falling into the water (e.g., fishing, canoeing, boating, etc.), as opposed to more extreme activities such as jet-skiing, water-skiing, etc., where more robust protection would be needed.

In response to this need, the objective of this work is to design and evaluate a proof-of-concept stoma-occlusion/breathing device that can be used to protect the airway of laryngectomees during unanticipated submersion in water, while allowing the individual to breathe after resurfacing should their stoma be below the waterline. This scenario is most likely to occur during participation in water activities. The manuscript is outlined as follows. § introduces the design objectives, § details the device design, and § describes the validation and performance of the device design. Finally, § is left for the conclusions.

## Design objectives

The design objectives of the device were determined by administering an institutional review board approved survey to participants of the Utica, NY Laryngectomy Support Group. Patient recruitment occurred through announcements at the monthly Utica, NY Laryngectomy Support Group. Survey's were completed at the conclusion of the meeting in a private setting (see S1 File). The Clarkson University Institutional Review Board approved the study (#20 − 30E). Written informed consent was obtained from all participants. The survey results, together with informal conversations, indicated the device should achieve the following objectives: (1) Automatically occlude water ingress into the stoma when submerged in water, without the need for user intervention and regardless of subject orientation when entering the water. (2) Allow comfortable respiration through through the device. (3) Not physically interfere with, or inhibit the preferred speech method of the user (tracheosophageal speech, esophageal speech, or electrolarynx speech). (4) Be comfortable and secure for the user to wear for the duration of participation in the activity (i.e., O(hrs) of usage time).

## Device design

The proposed device, referred to hereafter as the STORKEL (i.e., SToma-snORKEL), is comprised of three subcomponents, shown in Fig 1. They include, (1) the stoma attachment, (2)

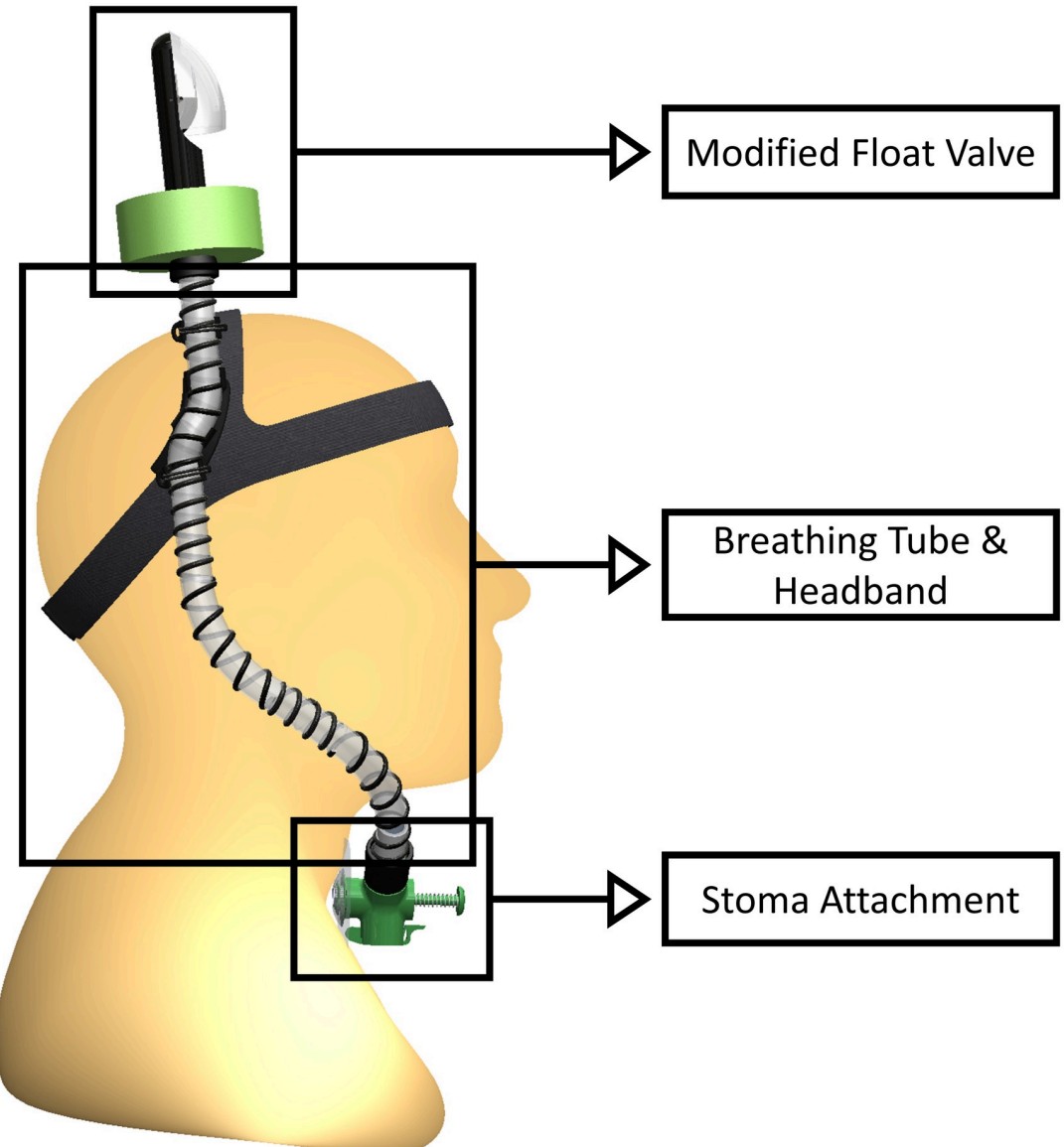

**Fig 1. The STORKEL and its subcomponents.**

breathing tube and headband, and (3) float valve. Together, these components fulfill all design objectives previously specified, providing a first, novel solution to a demonstrated need within the laryngectomy community. Due to the time constraints required to receive United States Food and Drug Administration (FDA) device approval, which is necessary prior to performing end-user evaluation of the device, the emphasis of the current work is to introduce the device, justify the design decisions, and report on proof-of-concept testing. It is emphasized that, as presented, the STORKEL is not recommended for current use as it has not received FDA approval. Furthermore, while the design approach of the device is evaluated, device safety is not proven in the current work. Consequently, the device should not be considered a suitable solution to prevent ingress of water through the stoma during participation in water activities until the appropriate approval and certification has been completed.

## Stoma attachment

The stoma attachment subcomponents (see Fig 1) enable a comfortable and secure attachment to the stoma, allow for TEP, esophageal, and electrolarynx speech, and maintain a waterproof interface between the pulmonary tract and the device during submersion.

Fig 2 shows a detailed view of the stoma attachment component, with subcomponents of the body, cap, insert, drain cap, and two-piece button valve delineated by dashed lines. The stoma attachment consists of an internal T-channel, where the leg of the T connects to the stoma so that the user breathes through the air channel in the stoma attachment. The upper branch splits, with the lower end terminating at the drain cap (see Fig 2(b)) while the other connects to the tube of the head band and breathing tube subcomponent (Fig 1).

The stoma attachment interfaces with the tracheostoma of the user via a soft, conical silicon pad that has a 1.40 cm(0.55 in) outer diameter, and a 1.17 cm (0.46 in) interior channel that aligns with the air channel of the stoma attachment. The exterior conical surface of the silicone pad has a slight convex shape to it. A small cut-out in the silicone pad mates with a flange on the stoma attachment to ensure the silicone pad stays in place while allowing it to be easily replaced. With the silicone pad positioned over the user's stoma, a square waterproof flexible polyethylene adhesive patch that measures 10.2 cm × 10.2 cm (4.0 in × 4.0 in) is attached to a small external flange over the stoma attachment insert, with the adhesive side facing the neck of the user. The adhesive patch has an ≈ 2.5 cm (1.0 in) hole through center through which the end of the silicone pad is inserted. With the silicone pad in a comfortable position, the adhesive is then attached to the neck of the user, sealing the airway with the stoma attachment and affixing the stoma attachment to the neck of the user.

The stoma attachment is small enough that it does not interfere with the use of an electorlarynx device for replacement speech, nor does it impede esophageal speech. To produce TES while wearing the stoma attachment, a user must be able to occlude the stoma so that air can be redirected from the trachea, through the TEP, and into the esophagus. This is accomplished via activation of a button valve that is connected to a silicone plunger via a 6.22 cm (0.25 in)

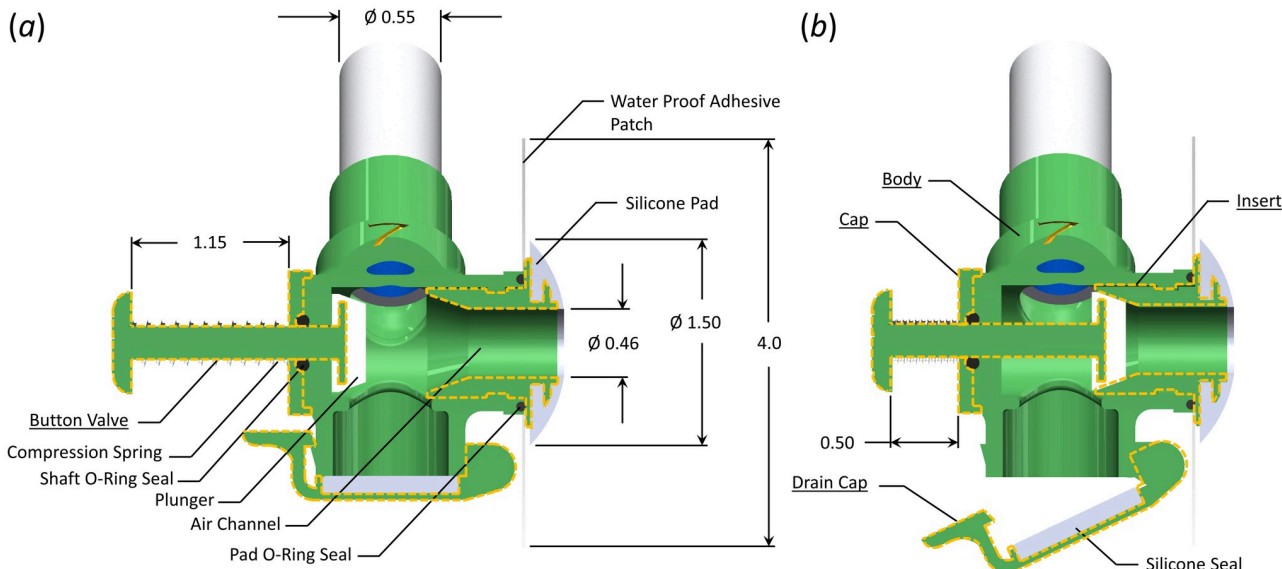

**Fig 2. Schematic of the stoma attachment.** (a) Orientation of the stoma attachment during breathing. (b) Orientation of the stoma attachment when the button valve has been depressed to enable tracheoesophageal speech. The drain cap is also open to show how accumulated water within the device can be drained. All dimensions are in inches.

diameter shaft. When depressed 1.65 cm (0.65 in) (see Fig 2(b)) the plunger depresses into a chamfer in the interior channel, sealing off the airway. A compression spring is positioned circumferentially around the shaft of the plunger, positioned between the button of the valve, and the body of the stoma attachment. When the button is released, the spring returns the plunger to the breathing position. The shaft of the button valve passes through an O-ring to maintain an airtight seal.

The drain cap is positioned at the inferior end of the stoma attachment, and moves about a hinge joint, with a clasp holding it closed. A silicone disk circumferentially seals the drain cap against the stoma attachment. In the event of a small amount of water accumulation in the device, the clasp can be quickly released to open the drain cap, drain the water, and then close it without removing the stoma attachment from the neck of the user. The drain cap is located at the lowest elevation point of the STORKEL to ensure that even if water does enter the device, it will collect within the drain cap, as opposed to entering the airway. When the individual is not at risk of drowning (e.g., while wearing the device on land) the flap can also be opened to eliminate any airflow resistance that occurs from breathing through the attached breathing tube and float valve.

Finally, the superiorly-positioned exit to the air channel is angled at $\approx 45°$ from vertical such that the circumferentially secured breathing tube exits the stoma attachment at a comfortable angle that prevents excessive bending of the breathing tube to conform to the head and neck geometry of the user.

The stoma attachment was printed as six separate pieces: body, cap, insert, drain cap, and two-piece button valve. Each of the pieces was printed on a Raise3D© Pro2 Plus printer with a 0.01 mm ($39.3 \times 10^{-6}$ in) layer build height and a 0.2 mm ($7.8 \times 10^{-3}$ in) nozzle diameter using 1.75 mm (0.059 in) diameter Raised3D© standard PLA filament. The stoma attachment is assembled by first seating the shaft o-ring into the interior cavity, and then securing it with the cap. The silicone plunger is then attached to the flange of the button valve. Next, the shaft of the button valve is inserted from the interior of the stoma attachment through the hole of the cap. The head of the button valve is then attached to the shaft. The silicone seal is affixed to the surface of the drain cap, and then the drain cap is attached to the bottom of the body by placing a pin through the hinge joint and affixing the clasp. Separately, the silicone pad is placed over the end flange on the insert, and the waterproof adhesive is affixed to the opposing face. Finally, the pad o-ring is placed in a groove in the body, and then the entire insert, with the attached waterproof adhesive and silicone pad, is inserted into the inner channel of the stoma attachment body. The insert has two internal J-channels positioned 180° relative to each other that are recessed into the external diameter of the insert. The mating interior surface of the body has two corresponding posts that protrude interiorly into the channel. This forms a quick-disconnect, similar to a BNC cable attachment, albeit inverted for an internal channel. When placing the insert inside the body, the pad o-ring is compressed sealing the interface, while providing enough elasticity that it holds the insert in place. The compression spring, the Shaft O-ring, and the Pad O-ring were all purchased commercially. Both the silicone pad, and the silicone seal on the drain cap were fabricated from custom printed molds using Smooth-On Dragon Skin™ silicone, which is approved for skin contact.

## Breathing tube

The breathing tubing connects the airway between the stoma attachment and the modified float valve. Both ends of the tube connect to the stoma attachment and modified float valve via a compression fit, rubber cuff with an outer diameter of 3.18 cm (1.25 in) that creates an airtight seal while still allowing disassembly to enable tube replacement, if desired. The interior of

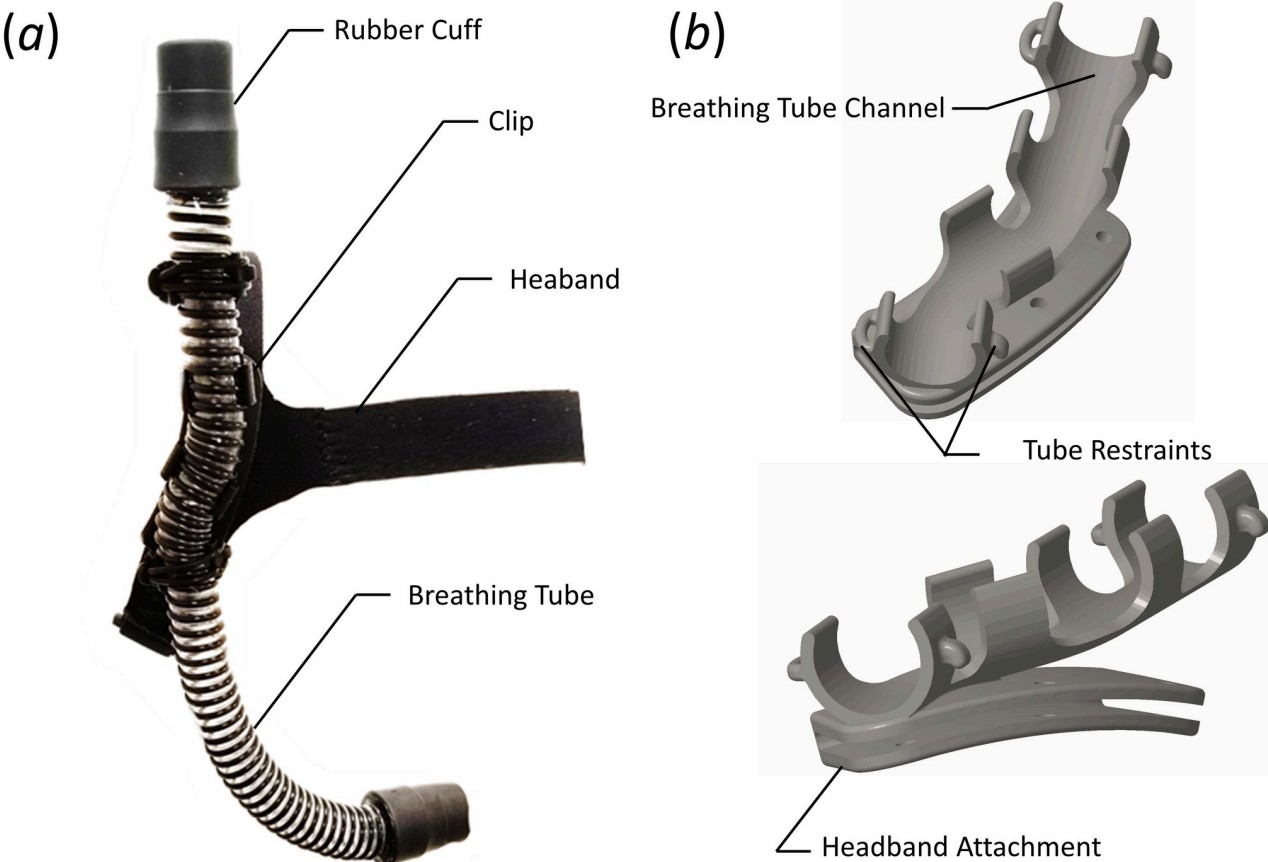

**Fig 3.** (a) Image of the assembled headband, breathing tube, and clip. (b) Perspective view of the clip.

the breathing tube is lined with two lengths of 5.84 cm (2.30 in) long, 3.0 cm (1.18 in) wide, and 0.3 cm (0.12 in) thick polyurethane water absorbent tape (by.RHO condensation tape) such that each length of tape wraps circumferentially around the inner diameter of the tube, with both strips placed adjacent to each other in the axial direction. The tubing attaches to the headband via a custom-designed clip. The headband, breathing tube, and the clip that connects them together are shown in Fig 3(a), with a detailed view of the clip shown in Fig 3(b).

The tubing (RespLabs® Medical Inc. CPAP hose) is comprised of a thin, flexible plastic tube with a wall thickness of $\approx$ 0.25 mm (0.01 in), and an internal diameter of $D$ = 1.83 cm (0.72 in). A plastic helical coil is bonded to the external diameter of the tube. The helical coil supports the plastic tube, producing a lightweight, flexible hose with a smooth internal diameter. This tubing was chosen due to its flexibility while resisting creasing and kinking, and its light weight. The tube is $\approx$ 31 cm (12 in) long, but can be adjusted as needed for each individual user. The specific length of the tube is determined by the user's size, and to ensure that there is at least 2.5 cm (1.0 in) of tubing between the cuff attached to the modified float valve and the distal end of the headband clip, as will be discussed later.

The diameter of the breathing tubing was chosen to ensure that the user could comfortably generate an adequate amount of air flow, which for the average human corresponds to $Q \approx$ 0.0014 m$^3$/s (85 L/min) at light to moderate levels of physical activity [34], with a respiratory rate of $\approx$ 10 − 15 breaths per minute. This results in a Reynolds number of Re = $(4\rho Q)/(\pi D\mu)$ $\approx$ 6, 500, where $\rho$ and $\mu$ are the density and dynamic viscosity of the fluid, respectively. The

Reynolds number provides a measure of the ratio of inertial to viscous effects in the flow. The Womersley number, $\alpha$, which provides a measure of puslatility to viscous effects within a pulsatile flow, is given as

$$\alpha = D\left(\frac{\omega\rho}{\mu}\right)^{\frac{1}{2}},\tag{1}$$

where $\omega$ is the angular frequency of the oscillations. Based on the aforementioned physiological measures of breathing the Womersley number can be computed as $\alpha \approx 5$. For reasonably low Womersley numbers, the ratio of oscillatory to steady flow impedance ($Z_o$ and $Z_s$, respectively) can be simplified as

$$\frac{Z_o}{Z_s} \approx 1 + \frac{i\alpha^2}{8},\tag{2}$$

where the impedance is defined as a phasor, such that

$$Z = \frac{\partial p/\partial z}{\hat{Q}},\tag{3}$$

with $z$ denoting the axial flow direction.

The steady state impedance can be estimated by solving the steady-flow energy equation for the pressure loss at the desired flow rate. The steady flow pressure loss through a system can be expressed as

$$\Delta p = \sum h_l = \left(\frac{8\rho Q^2}{\pi D^4}\right)\left[f\frac{L}{D} + \sum K_i\right],\tag{4}$$

where $f$ is the Darcy friction factor, and $K$ is the device loss coefficient. The first term on the right hand side comprises the major losses while the second term is the minor losses. Minor losses were included to account for one 90° degree juncture in the stoma attachment, two 90° degree bends in the breathing tube, and exit losses at the end of the snorkel valve. These values were computed as $K_{junc} = 1.5$, $2K_{bend} = 1.8$, and $K_{exit} = 1.0$, respectively. For the specified breathing tube diameter and flow rate, the steady state flow loss is $\Delta p \approx 83.5$ Pa. Expressing this as an impedance and solving for the impedance of the oscillatory component, $Z_o$, allows the magnitude ot he pulsatile pressure loss to be computed, which was found to be $\approx 275$ Pa. While this estimated value neglects losses due to flow curvature around bends in the tube, it provides a reasonable estimate of the anticipated pulmonary pressure needed to breathe through the STORKEL. In comparison, during comfortable breathing, pulmonary pressures of $\approx 200$–650 Pa are easily achievable [34]. Consequently, the anticipated pressure loss through the system is such that it is not anticipated to cause any respiratory distress during light activity. This conclusion was anecdotally validated by noting the ability to comfortably breathe through the device. Care should be taken, however, in the use of the device with elderly individuals with compromised pulmonary function.

## Headband

The headband ensures the STORKEL remains securely attached to the head and properly positioned during submersion under water. The head band is a custom-design that is fabricated from 3 mm (0.12 in) thick neoprene fabric. Neoprene was chosen because it is comfortable, provides some stretch, and will not deteriorate in water. The design uses a Y-yoke that fits around the forehead, and across the crown of the skull, while a single band is positioned to fit under the base of the skull. The strap at the base of the skull is split, and has a Velcro® fastener

that connects both ends together, allowing adjustment to accommodate a wide range of head sizes. Ribbons of Ethylene-vinyl acetate are applied to the inner surfaces of the headband that contact the head to provide additional grip so that the headband does not slide around on the user.

The breathing tube is attached to the headband via a custom-designed clip. The clip consists of a channel that the breathing tube snaps into, with four retaining rings (see Fig 3(b)). The first and last retaining rings have a hook and loop on the opposing edges. A rubber ring is passed through the loop, stretched over the top of the tube, and then placed over the hook, securing the tube to the headband attachment. The headband attachment (see Fig 3(b)) slides over the neoprene fabric of the headband, and is positioned behind the right ear. Note the contour of the headband attachment is such that is follows the contour of the skull (see the lower panel of Fig 3(b)), while the channel for the tube continues in a vertical plane when worn by the user. This directs the tube away from the head so that the user has the option to don headwear with the device. The tube length can be adjusted based on the physiology of the user, although a length of 35 cm (14 in) was found to be acceptable for most users. Once properly adjusted the clip is secured to the neoprene headband with cyanoacrylate glue.

## Float valve

The float valve, shown in Fig 4, is affixed to the rubber cuff at the superior end of the breathing tube, and ensures that regardless of angle of entry or method of immersion in water, the breathing tube automatically seals off the airway while it is submerged below water, and then automatically opens upon resurfacing above the water. The modified float valve consists of an Aegend© Dry Snorkel valve fitted to the breathing tube. A cylindrical flotation ring of polyethylene foam is affixed around the rubber cuff, at the base of the dry snorkel valve. The flotation ring has an inner diameter of 3.2 cm (1.25 in), an outer diameter of 11.4 cm (4.5 in), and is 3.8 cm (1.5 in) thick. The breathing tube length is adjusted such that the position of the bottom of the flotation ring relative to the exit of the breathing tube channel in the headband clip is adjustable (denoted in Fig 4 as $h$). As will be discussed later, this distance was optimized to ensure proper performance of the valve, while also maximizing comfort for the user.

The float valve uses competing buoyancy and gravitational forces about a hinged joint to remain closed when entering the water. A schematic showing the actuation of the valve is presented in Fig 5. The valve consists of a hollow rotational stopper that rotates about a hinge joint. A flat flapper is attached to the rotational stopper, which can seal the exit area through which air is breathed. The weight of the rotational stopper and the volume of the interior cavity are designed such that the resulting gravitational and buoyancy forces that arises during submersion are of the same magnitude. When in a vertical orientation above the waterline, gravity acting on the rotational stopper causes it to rotate around the hinge joint and remain in the open position (solid line in Fig 5(a)). As the modified float valve enters the water in this same orientation, a buoyancy force is created by the hollow rotational stopper. As shown in Fig 5(a), the line of action ($x_b$) of the buoyancy force ($F_b$) about the hinge point is greater than the line of action ($x_g$) for the gravity force ($F_g$). This causes the rotational stopper to rise with the water level, and seal off the airway before it can become submerged in the water. As shown by the dashed lines, when the rotational stopper is closed, the location of the gravitational and buoyancy forces are still such that the device remains closed. In an inverted orientation (solid line in Fig 5(b)) the gravity force causes the valve to close, prior to entering the water. As the water level rises, the line of action ($x_g$) of the gravity force remains greater than the line-of-action of the buoyancy force ($x$b) and so the valve remains closed. This passive activation ensures the

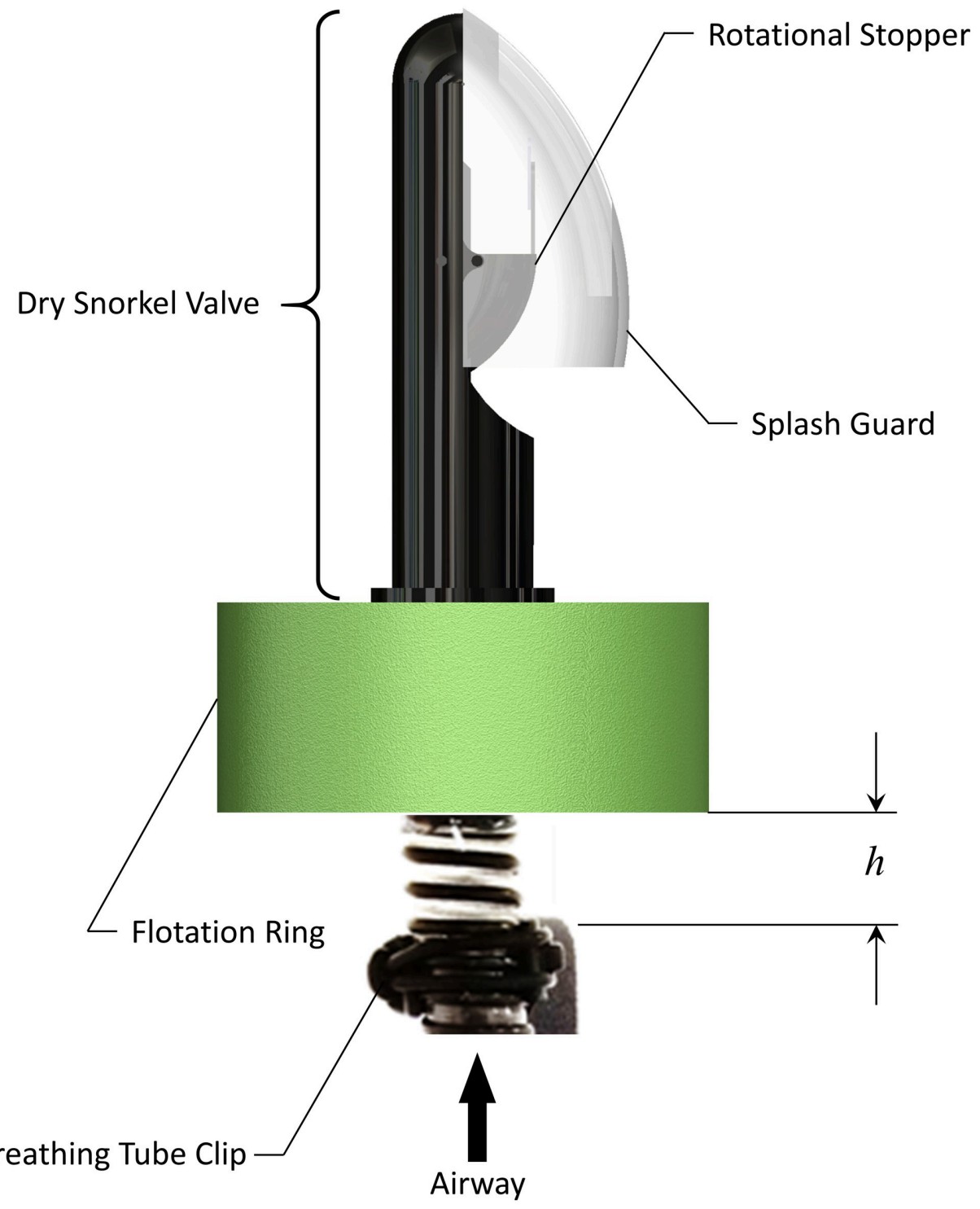

**Fig 4. Components of the float valve.**

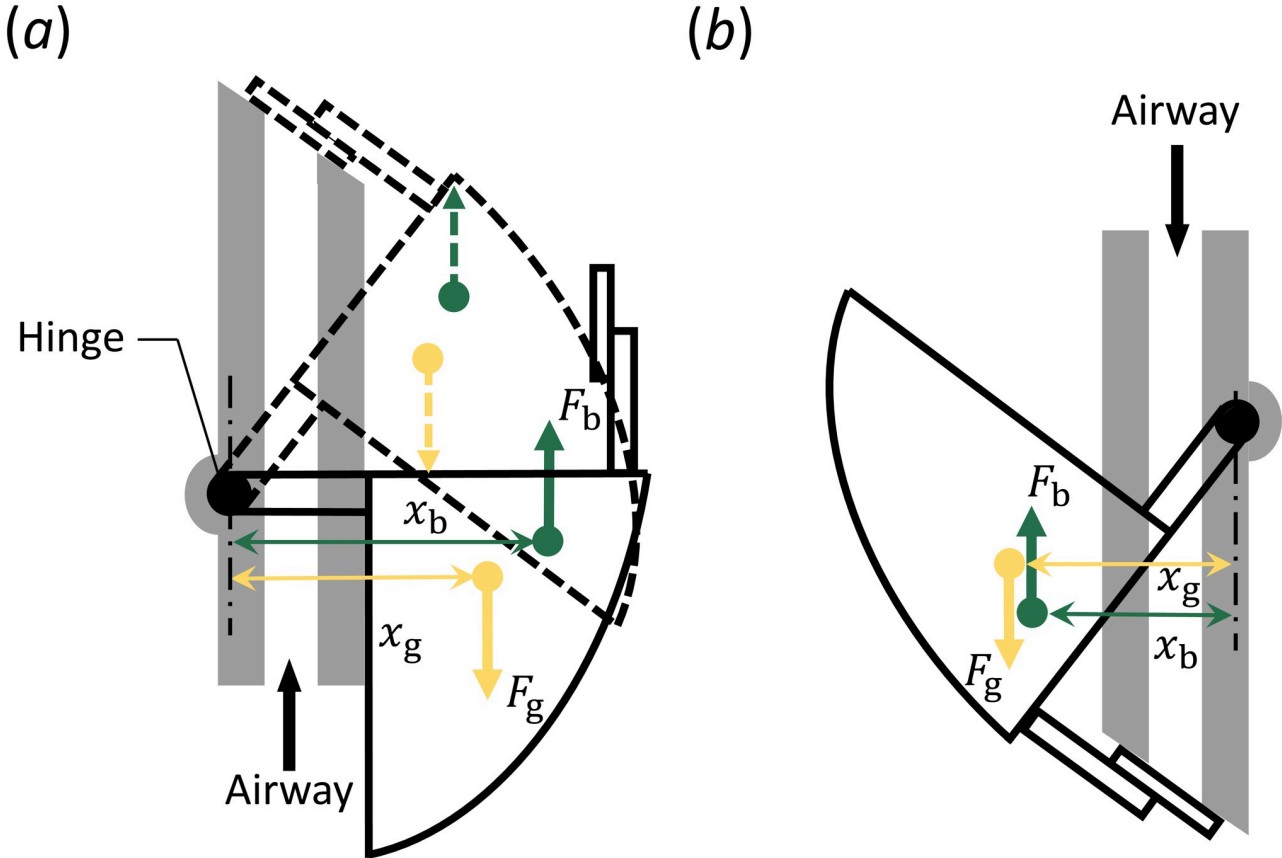

**Fig 5. Float valve actuation.** (a) Activation of the valve when in a vertical position prior to being submerged (solid line), and during submersion (dashed line). (b) Actuation of the valve when inverted, regardless of whether it is submerged or not.

breathing tube is sealed off prior to water entry, remains closed when submerged, and automatically opens upon resurfacing in the upright position.

**Flotation ring.** Because the hinge joint in the float valve only enables rotation about one plane, it does not function correctly if the trajectory of water entry is aligned with the axis of the hinge joint, or a significant component of it. To address this, a flotation ring was added to the breathing tube. As previously mentioned, the flexible breathing tube extends above the headband clip, where it is secured to the head. This length provides mobility in the tube ensuring that as the float valve enters the water from any angle, the buoyancy of the flotation ring will orient it into a nearly vertical position. This then allows the hinge joint to operate as designed, protecting the airway. However, an excessive length of the breathing tube could be considered an annoyance by the user.

## Device evaluation

To ensure adequate performance of the float valve and flotation ring, the performance was evaluated through submersion tests. The breathing tube and modified float valve were secured to a standard brick measuring 20.3 cm × 9.1 cm × 5.8 cm (8.0 in × 3.6 in × 2.3 in) with a mass of ≈ 2.3 kg (5.1 lb), with the breathing tube capped on the end opposite of the modified float valve. The breathing tube and the modified float valve were placed on the face of the brick, with the flotation ring and modified float valve extending past the end, by an adjustable

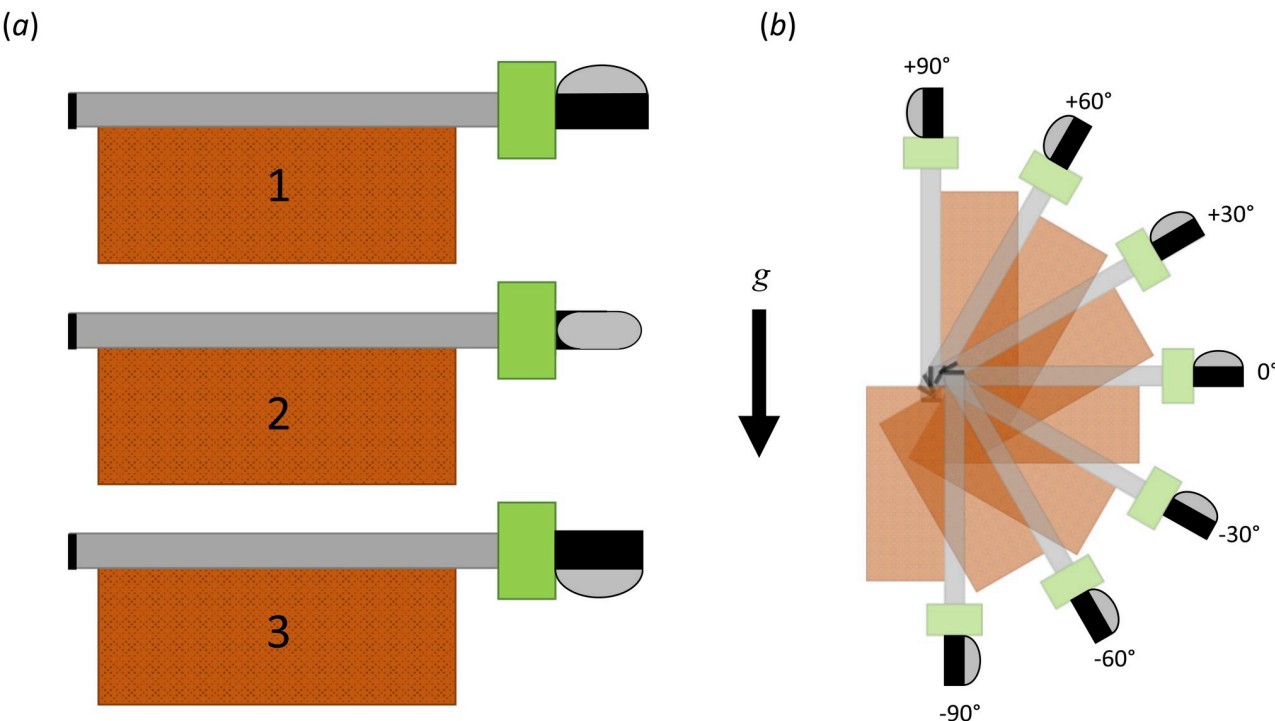

**Fig 6. Orientation of the STORKEL relative to the angle of entry for water submersion tests, indicated relative to gravity, *g*.** (a) Investigated water entry orientations of the float valve for the plane parallel to the water surface. (b) Investigated water entry orientations of the modified float valve relative to the plane of the water surface.

distance, *h*, as shown in Fig 6. This distance was investigated for values of $h = 0.0$ cm (0.0 in), $h = 2.54$ cm (1.0 in), and $h = 5.08$ cm (2.0 in). The angle of orientation of the modified float valve relative to a water entry parallel to the plane of the water was also adjusted, as shown in Fig 6(a). The orientation of the float valve was varied so that the splash cover was oriented facing away from the plane of the water (Fig 6(a.1)), parallel to the water (Fig 6(a.2)), and towards the water (Fig 6(a.3)). Finally, the performance of the modified float valve was tested as a function of water entry angle in 15 degree increments, as shown in Fig 6(b), and for each orientation of the float valve relative to the plane of the water. For each orientation, the device was dropped from a height of 1.5 m (5 ft) into a large body of water (i.e., the University swimming pool). For each run, the amount of water that entered the device through the modified float valve was collected and measured.

The volume of water that entered through the modified float valve during submersion tests is plotted in Fig 7 as a function of the angle of entry for the three spacings between the flotation ring and the attachment point (*h*). The values reported are averaged at each entry angle across the three orientations (see Fig 6). The mean value for each distance, averaged across all entry angles are, 9.05 mL (0.55 in³), 2.94 mL (0.18 in³), and 2.80 mL (0.17 in³) for $h = 0.0$ cm (0.0 in), 2.54 cm (1.0 in), and 5.08 cm (2.0 in), respectively. As the spacing between the flotation ring, and the point of fixation (the head band clip for the physiological scenario) increased, the amount of water that enters the modified float valve decreased. This is because the flexibility of the breathing tube allows the flotation ring to more easily orient the float valve into the vertical position before it is submerged, due to its buoyancy. This minimizes the likelihood that the float valve will be submerged when the hinge joint is perpendicular to the surface of the water; the orientation whereby the hinge joint will not function to occlude water entry. An offset

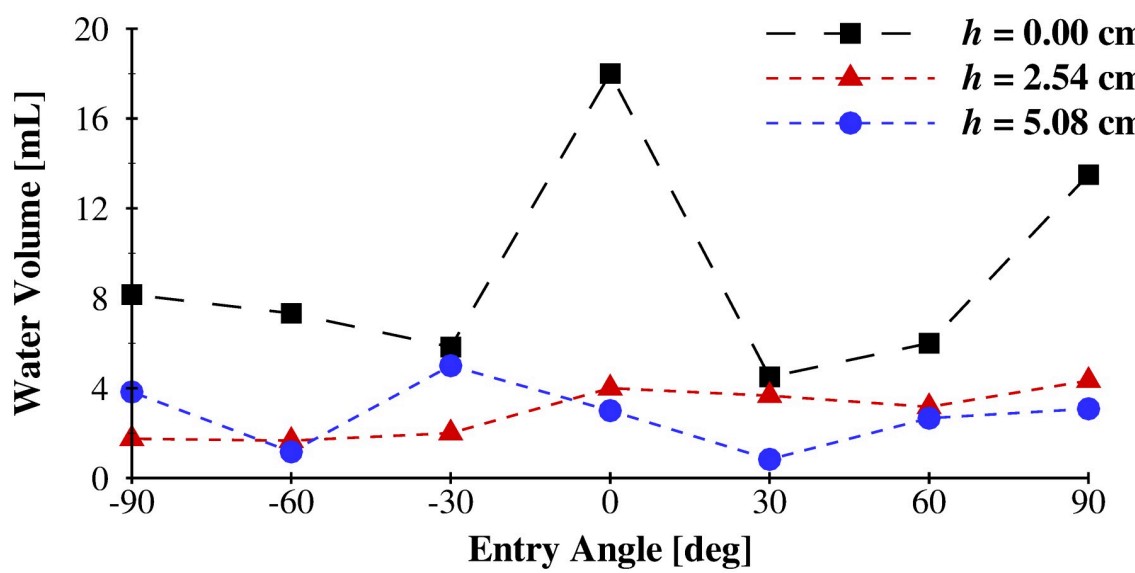

**Fig 7. Volume of water that entered the STORKEL as a function of flotation ring location (*h* in Fig 4), and water entry angle for orientation 2 (see Fig 6).**

distance of $h$ = 2.54 cm (1.0 in) was selected as the ideal distance between the bottom of the flotation ring, and the attachment point on the head band clip, which minimized water ingestion while maximizing user comfort, as it was found that due to the flexibility of the breathing tube, an excessively long tube length gave rise to excessive and annoying bobbing of the modified float valve during use.

Finally, the entire STORKEL device was evaluated by placing it on the head of one of the investigators (albeit with the stoma attachment simply capped off, as opposed to connected to a laryngectomee), for which IRB approval was not required, and informed consent was provided. The user entered the water in orientations that were representative of real life conditions. Standing at the edge of the pool ($\approx$ 15.2 cm (6.0 in) above the water surface) the user tested three entry orientations: jumping in feet first, falling in face down, and falling in face up (on their back). Each trial was repeated 5 times. The average amount of of water that was collected from the drain cap (i.e., that had entered the STORKEL) for the three water entry orientations was 0.0 mL($0.0in^3$), 1.5 mL ($0.23in^3$), and 1.0 mL($0.15in^3$), and for the feet first, face down, and face up water entry orientations, respectively. These amounts of water entering the STORKEL were deemed acceptable as they are small enough they would not pose any significantly harmful effects. Additionally, the orientation of the drain cap, which is placed below the entrance to the stoma, increases the likelihood that even in the event of water entering the STORKEL, it will collect in the drain cap, as opposed to entering the airway.

Note that proof-of-concept evaluation of the device was not performed with laryngectomees. This may introduce additional circumstances that have not been explicitly addressed herein. For example, the integrity of the waterproof adhesive that connects the stoma attachment to the skin of the neck was not explicitly tested. While the adhesive is specifically designed to be waterproof, the effectiveness of the adhesive when coupled to the device remains to be evaluated. This is especially pertinent to laryngectomees who may suffer from skin degradation and localized sensitivity/inflammation due to radiation treatments for head and neck cancer. Similarly, varying morphometries of the stoma opening may influence fit, comfort and function of the stoma attachment. In addition, flow resistances through the device were determined to be suitable for normal breathing conditions. However,

laryngectomees often have co-morbidities that influence pulmonary function. The influence of reduced lung capacity on the comfort and effectiveness of the device also remains to be evaluated. These potential complications highlight the need to consider the current device a proof-of-concept solution, and also identify possible challenges to be addressed in the future. Finally, as recommended, when participating in any water activity, use of a life preserver would severely diminish the likelihood of complete submersion of the STORKEL during accidental water entry, and would further decrease the likelihood of water entry into the pulmonary tract of a laryngectomee.

## Conclusions

A proof-of-concept SToma snORKEL (STORKEL) has been designed and tested as an assistive device to enable laryngectomees to safely participate in water activities where accidental, sudden immersion poses a life-threatening risk. A novel approach has been outlined that utilizes a silicone and adhesive stoma interface connected to a breathing tube, which seals around the stoma and facilitates breathing through a flexible tube. The breathing tube is routed from the stoma, through a head band, with the opening located above the top of the head of the user, similar to a snorkel. The opening of the tube is connected to a mfloat valve that passively occludes the airway when the opening is submerged under water, and then automatically opens again upon surfacing. The tubing was sized to allow comfortable breathing during use. A minimal amount of water was found to bypass the modified stoma valve during submersion tests, which was addressed by lining a portion of the interior circumference of the breathing tube with water absorbent tape to collect the small amount of water that may enter the breathing tube. Final testing of the device in scenarios indicative of real-life usage demonstrated that at most, a minimal amount of water ($< 1.5$ mL) was able to enter the STORKEL and bypass the water absorbent tape, thereby potentially entering the airway of the user. It is emphasized that because the device does not have FDA approval, and the effectiveness of it when in use has not been proven, it is not currently suitable for personal use.

## Supporting information

**S1 File. Laryngectomee survey.** Survey administered to the study participants.
(PDF)

## Author Contributions

**Conceptualization:** Samantha K. Denning, Michael A. Valleau, William J. Pelowski, Claire M. Chaisson, Kelli E. Grimes, Byron D. Erath.

**Formal analysis:** Samantha K. Denning, Michael A. Valleau, William J. Pelowski, Claire M. Chaisson, Kelli E. Grimes, Byron D. Erath.

**Funding acquisition:** Byron D. Erath.

**Investigation:** Samantha K. Denning, Michael A. Valleau, William J. Pelowski, Claire M. Chaisson, Kelli E. Grimes, Byron D. Erath.

**Methodology:** Samantha K. Denning, Michael A. Valleau, William J. Pelowski, Claire M. Chaisson, Kelli E. Grimes, Byron D. Erath.

**Project administration:** Byron D. Erath.

**Supervision:** Byron D. Erath.

**Writing – original draft:** Samantha K. Denning, Byron D. Erath.

**Writing – review & editing:** Samantha K. Denning, Byron D. Erath.

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
