## [Decision Letter · Decision Letter 0]

22 Jul 2021

PONE-D-21-14821

An automatic water-occluding device to enable laryngectomee participation in water activities

PLOS ONE

Dear Dr. Erath,

Thank you for submitting your manuscript to PLOS ONE. After careful consideration, we feel that it has merit but does not fully meet PLOS ONE’s publication criteria as it currently stands. Therefore, we invite you to submit a revised version of the manuscript that addresses the points raised during the review process.

ACADEMIC EDITOR:  I agree with the reviewers. Please address their concerns and revise the paper to be a proof of concept with a warning that it does not prove safety. A warning is needed that the device is not FDA (or similar) approved and further safety data is needed before it can be recommended for patients.

We look forward to receiving your revised manuscript.

Kind regards,

Peter Dziegielewski, MD, FRCSC

Academic Editor

PLOS ONE

Journal Requirements:

2. Please provide additional details regarding participant consent. In the ethics statement in the Methods and online submission information, please ensure that you have specified whether consent was informed.

5. We note that Figures 1, 3, 4 and 5 in your submission contain copyrighted images. All PLOS content is published under the Creative Commons Attribution License (CC BY 4.0), which means that the manuscript, images, and Supporting Information files will be freely available online, and any third party is permitted to access, download, copy, distribute, and use these materials in any way, even commercially, with proper attribution. For more information, see our copyright guidelines: http://journals.plos.org/plosone/s/licenses-and-copyright.

a. You may seek permission from the original copyright holder of igures 1, 3, 4 and 5 to publish the content specifically under the CC BY 4.0 license. 

Additional Editor Comments:

Thank you for your submission. This is an interesting paper about a potentially helpful device in laryngectomee patients. However, I agree with the reviewers that this study doe not prove safety. Publishing it as such may put patient's in harm's way and could be dangerous. This needs to be revised to be a proof of concept paper and there needs to be a warning that this study does not prove safety or recommend the use of the device until further investigation is carried out and FDA approval (or similar) is passed.

Reviewers' comments:

Reviewer's Responses to Questions

**Comments to the Author**

1. Is the manuscript technically sound, and do the data support the conclusions?

Reviewer #1: Partly

Reviewer #2: Yes

Reviewer #3: Partly

2. Has the statistical analysis been performed appropriately and rigorously? 

Reviewer #1: N/A

Reviewer #2: N/A

Reviewer #3: I Don't Know

3. Have the authors made all data underlying the findings in their manuscript fully available?

Reviewer #1: Yes

Reviewer #2: Yes

Reviewer #3: Yes

4. Is the manuscript presented in an intelligible fashion and written in standard English?

Reviewer #1: Yes

Reviewer #2: Yes

Reviewer #3: Yes

5. Review Comments to the Author

Reviewer #1: In their manuscript, the authors describe their novel device designed to allow laryngectomy patients the ability to engage in water activities.

There are some unneccessary commercial items listed around line 47.

the largest issue is the ethics of this study. the authors do not differentiate the risk of the various water sports (ie fishing from a dock vs paddleboarding or kayaking).

the itegrity of the "waterproof" adhesive is not addressed.

this may be better suited as a proof of concept manuscript with explicit discussion that is not meant for actual patient use at this time

Reviewer #2: A well written manuscript about a new device for larygectomees' patients. As device was only tested in normal people, it remains untested in real patients and there may be unforeseen issues. I suggest to add these into the discussion. What are the unforeseen circumstances that may arise and what measures can be instituted to overcome them?

Reviewer #3: The aim of this paper was to establish proof of concept for a tracheostoma occluding device for use in the event of unanticipated immersion in water. With such a safety device, individuals who have undergone total laryngectomy may be more willing to participate in various recreational water activities, thus improving their quality of life, which is significantly compromised after such surgery. For those who already participate in water activities, this type of device may increase survival rates of those who experience accidental submersion. The design was based on survey responses of individuals who have undergone total laryngectomy from a local support group. Based on that feedback, design objectives included automatic water occlusion, comfortable respiration, compatibility with alaryngeal speech including use of a tracheoesophageal voice prosthesis, and comfort for long wear time.

In reference to Line [8] of the Introduction, bilateral vocal fold paralysis and intractable aspiration are not causes of laryngeal dysfunction, but results. Cancer, trauma, or neurological insult are common etiologies of laryngeal impairment. In line [15], the authors go on to state that the primary function of the larynx is speech production. Many would argue that the primary function is airway protection, which would support the premise for the device.

The design and thus the name of the device is a bit confusing. SToma-snORKEL (STORKEL) implies a breathing device, not an arresting device. When the airway is submerged in water, the airway is closed on multiple levels, e.g., lips close, vocal folds adduct, diaphragm and lungs stop expanding, until the airway (typically the nose or mouth) has surfaced. For laryngectomees, the problem is twofold. First, the airway of a laryngectomee is redirected to the tracheostoma, and the vocal folds are no longer present to provide protection. Second, the stoma is positioned distally in the neck, which will often remain submerged when an individual is in deep water, unless they can float on their back and allow the stoma to rise above the surface.

Based on that premise, one can see why a "snorkel" type device would be advantageous, to enable respiration, should the stoma remain submerged for an extended period of time. This concept could be highlighted, especially in the event that a life preserver may not provide enough buoyancy to keep the stoma above the waterline.

While the occluding feature of the float valve was intensively assessed, the occlusion at the level of the stoma was not evaluated. If the main purpose of the device is to protect the stoma, then this should be assessed. Individuals who have undergone total laryngectomy often experience suboptimal baseplate seals due to sensitive skin and stoma landscape. Individuals who use a TEP experience additional challenges of excess pressure exerted on the baseplate when they occlude their stomas to generate TEP speech, thus often "break the seal." A broken seal while using this device would be life threatening. This could be easily tested on several laryngectomees (with IRB approval) by just having them speak and cough with the device on, and assessing if the seal stays intact.

The additional limitations and additional questions for future testing and design modification are suggested. In addition, did the authors present their design to the original laryngectomees who completed the design survey?

It is commendable that the needs of individuals who have undergone total laryngectomy have caught the attention of mechanical engineers. Collaborating with clinicians in the field of head and neck surgery, and speech-language pathology may all the more expedite the effort.

6. PLOS authors have the option to publish the peer review history of their article (what does this mean?). If published, this will include your full peer review and any attached files.

Reviewer #1: No

Reviewer #2: **Yes: **Professor Dr Baharudin Abdullah

Reviewer #3: No

---

## [Decision Letter · Decision Letter 1]

2 Sep 2021

An automatic water-occluding device to enable laryngectomee participation in water activities

PONE-D-21-14821R1

Dear Dr. Erath,

We’re pleased to inform you that your manuscript has been judged scientifically suitable for publication and will be formally accepted for publication once it meets all outstanding technical requirements.

Kind regards,

Peter Dziegielewski, MD, FRCSC

Academic Editor

PLOS ONE

Additional Editor Comments (optional):

Reviewers' comments:

Reviewer's Responses to Questions

**Comments to the Author**

1. If the authors have adequately addressed your comments raised in a previous round of review and you feel that this manuscript is now acceptable for publication, you may indicate that here to bypass the “Comments to the Author” section, enter your conflict of interest statement in the “Confidential to Editor” section, and submit your "Accept" recommendation.

Reviewer #1: All comments have been addressed

Reviewer #3: All comments have been addressed

2. Is the manuscript technically sound, and do the data support the conclusions?

Reviewer #1: Yes

Reviewer #3: Yes

3. Has the statistical analysis been performed appropriately and rigorously? 

Reviewer #1: N/A

Reviewer #3: I Don't Know

4. Have the authors made all data underlying the findings in their manuscript fully available?

Reviewer #1: Yes

Reviewer #3: Yes

5. Is the manuscript presented in an intelligible fashion and written in standard English?

Reviewer #1: Yes

Reviewer #3: Yes

6. Review Comments to the Author

Reviewer #1: (No Response)

Reviewer #3: The authors have adequately addressed my comments raised in a previous round of review. Specifically, that this is a proof of concept study, testing was not completed on individuals who have undergone laryngectomy, and the initial source of drowning is submersion of the stoma.

7. PLOS authors have the option to publish the peer review history of their article (what does this mean?). If published, this will include your full peer review and any attached files.

Reviewer #1: No

Reviewer #3: No

---

## [Editor Report · Acceptance letter]

3 Sep 2021

PONE-D-21-14821R1 

An automatic water-occluding device to enable laryngectomee participation in water activities  

Dear Dr. Erath:

I'm pleased to inform you that your manuscript has been deemed suitable for publication in PLOS ONE. Congratulations! Your manuscript is now with our production department. 

Kind regards, 

on behalf of

Dr. Peter Dziegielewski 

Academic Editor

PLOS ONE